# A Bacteriophage Microgel Effectively Treats the Multidrug-Resistant *Acinetobacter baumannii* Bacterial Infections in Burn Wounds

**DOI:** 10.3390/ph16070942

**Published:** 2023-06-29

**Authors:** Deepa Dehari, Aiswarya Chaudhuri, Dulla Naveen Kumar, Rohit Patil, Mayank Gangwar, Sonam Rastogi, Dinesh Kumar, Gopal Nath, Ashish Kumar Agrawal

**Affiliations:** 1Department of Pharmaceutical Engineering and Technology, Indian Institute of Technology (BHU), Varanasi 221005, India; deepa.dehari.rs.phe18@itbhu.ac.in (D.D.); aiswaryachaudhuri.rs.phe20@itbhu.ac.in (A.C.); dullanaveenkr.rs.phe20@itbhu.ac.in (D.N.K.); patilrohitpravin.phe21@itbhu.ac.in (R.P.); dinesh.phe@itbhu.ac.in (D.K.); 2Department of Microbiology, Institute of Medical Science, Banaras Hindu University, Varanasi 221005, India

**Keywords:** antimicrobial resistance, bacteriophage, multidrug resistance, wound infection, bacterial biofilm

## Abstract

Multidrug-resistant (MDR) *Acinetobacter baumannii* (*A. baumannii*) is one of the major pathogens present in burn wound infections. Biofilm formation makes it further challenging to treat with clinically available antibiotics. In the current work, we isolated the *A. baumannii*-specific bacteriophages (BPABΦ1), loaded into the chitosan microparticles followed by dispersion in gel, and evaluated therapeutic efficacy against MDR *A. baumannii* clinical strains. Isolated BPABΦ1 were found to belong to the *Corticoviridae* family, with burst size 102.12 ± 2.65 PFUs per infected host cell. The BPABΦ1 loaded chitosan microparticles were evaluated for quality attributes viz. size, PDI, surface morphology, in vitro release, etc. The developed formulation exhibited excellent antibiofilm eradication potential in vitro and effective wound healing after topical application.

## 1. Introduction

*A. baumannii* is a Gram-negative bacterium exhibiting multidrug resistance (MDR) to all the major antibiotics, viz. carbapenems, aminoglycoside, colistin, β-lactamase, etc. *A. baumannii* is also well known for biofilm formation and strong environmental adaptability, which collectively make difficult effective treatment, leading to death in some cases. MDR *A. baumannii* bacterial infection is more common in patients with underlying diseases (e.g., ventilator-associated pneumonia, urinary tract infections, and bacteremia) or therapies that require invasive procedures. *A. baumannii* has been observed as an extensive pathogen found in combat-associated wounds [1]. The end of an “antibiotic era” may soon become a reality due to the growing incidence of antibiotic resistance, even against clinically available newly discovered antibiotics. The United Nations has already identified MDR bacterial infections as a global challenge and predicted nearly 10 million deaths by 2050 due to this [2].

Bacteriophages (BPs) (viruses that infect the bacteria and cause their lysis) are considered one of the promising alternative therapies for MDR bacterial infections designed to effectively reduce the bacterial loads within the infected region [3]. They have a unique characteristic of bactericidal activity, increased specificity, less inherent toxicity, absence of cross-resistance with antibiotic classes, and self-multiplication in the existence of the bacterial host. Unlike antibiotics, bacteriophages overlook the commensal microbiota due to their strain-specific property. It was further observed that bacteriophage work against both Gram-positive and Gram-negative bacteria. Bacteriophages can be rapidly isolated due to their ubiquitous nature and are abundant in every ecological niche, thereby decreasing the development cost compared to antibiotics [4]. Clinical implementation of phage therapy in Western countries still faces major obstacles, especially regulatory issues related to poor quality and safety guidelines, instability of phage during formulation, tedious method of phage quantification, different efficacy against biofilm, evaluation of resistance, and weak regulatory frameworks [5].

There has been a rise in interest in studying the effectiveness of BPs in combination with antibiotics. There are some previously published reports that have investigated the combination of phages with antibiotics. Manohar et al. (2022) evaluated the synergistic effect of the MRM57 phase in combination with antibiotics (carbenicillin, cefotaxime, colistin, meropenem, etc.) for the elimination of *Citrobacter amalonaticus*. When carbenicillin (32 g/mL) was administered at 1/4 of its minimum inhibitory concentration (MIC) in conjunction with Citrophage at 10^6^ (PFU/mL), a six-log reduction was observed in comparison to the use of antibiotics alone [6]. Care products for burn wounds and their active ingredients usually exhibit high acidity that can negatively affect the activity of phages in wounds [7].

The topical application of bacteriophages has been proclaimed in various applications such as infected burn-mediated ulcers and other chronically infected wounds. It was discovered that the efficacy of the bacteriophage treatment is mainly determined by bacteriophage concentration, implying that increasing MOI (multiplicity of infection) reduces the bacterial population and enhances phage therapy. In the case of an infected epidermis, the infection site is quickly colonized by bacteria, forming biofilms that cause resistance to one or multiple antibiotics. Such a situation can be overcome by applying bacteriophages, as supported by Kumari S et al. 2011, where the full-thickness wounds with *Klebsiella pneumoniae* were topically treated with silver nitrate, gentamicin, and phage Kpn5. It was demonstrated that an increased antibacterial efficacy was observed by Kpn5, followed by silver nitrate (0.5%) and gentamicin (1000 mg) [8]. Despite such advantages, protecting them from harsh environments like pH, temperature, and solvents while maintaining bacteriolytic activity is a big challenge. Moreover, poor diffusion and low penetration ability of bacteriophages make difficult for them to reach the infection sites. The emergence of alternative phage-based formulations, such as microparticles and nanoparticles, can overturn such drawbacks, paving a reasonable way for improved clinical outcomes. The encapsulation of bacteriophages within the nanoformulations improves their bioavailability, inhibits their inactivation due to enzymatic activity, offers a shielding effect against neutralizing antibodies, bypasses the elimination by RES, increases the long-term storage stability, and provides controlled phage release [9].

In a study, Ilomuanya et al. (2022) reported the formulation of the Acinetobacter baumannii phage cocktail loaded with chitosan microparticles for treating MDR wound infection. They have isolated the bacteriophages, namely, ϕAB140 and ϕAB150 against Acinetobacter baumannii simultaneously loaded into the chitosan microparticle for managing diabetic wound infection. The in vivo wound healing study demonstrated that the formulation produced maximum wound healing in 7 days. Additionally, it was found that the optimized formulation was nontoxic and safe for administration [10]. In another study, Ma et al. (2008) developed bacteriophage Felix O1-loaded chitosan alginate micro-formulations for oral delivery. The stability study suggested that free bacteriophage was extremely sensitive to the acidic environment and ultimately died within 5 min at 3.7 pH. However, the bacteriophage encapsulated within had good stability even at 2.4 pH. Further, an in vitro release study demonstrated that the bacteriophage was completely released in 6 h at pH 6.8 in simulated intestinal fluids. Hence, a developed formulation can deliver the encapsulated bacteriophage to the gastrointestinal tract for therapeutic action [11]. Weppelmann et al. (2020) recently designed the chitosan microparticle for the vibriocidal activity toward MDR cholera infections. It was observed that chitosan microparticles of weight above 0.1% efficiently killed the Vibrio cholera [12]. Moreover, Park et al. (2002) demonstrated that chitosan microparticles controlled the drug delivery of the loaded felodipine as the sodium TTP concentration increased, resulting in a slower release of the felodipine.

Additionally, as the molecular weight of the chitosan decreased, a faster release rate was obtained [13]. A few years back, Sette-de-Souza et al. (2020) reported the antibacterial activity of Schinopsis brasiliensis extract loaded with chitosan microparticles for infectious disease. Microparticles were found to be safe based on the low hemolytic potential assay. Antimicrobial tests showed that microparticles could stop the growth of *E. faecalis* ATCC as well as the development of biofilms [14].

In the present study, we have developed BP-loaded chitosan microparticles laden gel for treating MDR biofilm-mediated burn wound bacterial infections. Chitosan-based microparticles have been employed as a carrier of small molecules, bacteriophages, and biomacromolecules for treating MDR bacterial infection. As a biodegradable, biocompatible, and nontoxic polymer, chitosan offers excellent potential in pharmaceutical applications. [10]. However, chitosan has poor water solubility, so acetic acid may be required to dissolve it, but this can reduce bacteriophage titer. Therefore, we have used chitosan of a water-soluble grade to minimize this risk.

In addition to chitosan, we have also used trehalose to protect the capsid of BP against desiccation and thermal stress, which was achieved by creating a direct hydrogen bond with the capsid protein, thereby providing structural and conformational stability [15]. Further, to provide a local action, the developed chitosan microparticles were incorporated into a gel, which was accomplished using SEPINEO™ P 600 and glycerol (to prevent damage of bacteriophage membrane and moisturize the skin) which improves the stability, applicability, and texture of the final formulation [16]. Henceforth, in the current study, we have developed the BP-loaded microparticle-laden gel to assess the therapeutic effect of lytic bacteriophage formulation on the biofilm-associated infection caused by *A. baumannii* bacteria.

## 2. Results

### 2.1. Bacterial Strain Collection and Antibiotic Susceptibility Test

The obtained *A. baumannii* samples were sub-cultured and maintained on nutrient agar slants at 4.0 ± 1 °C for future use. *A. baumannii* was resistant to piperacillin/tazobactam, cefepime, co-trimoxazole, amikacin, ceftriaxone, gentamicin, cefalexin, ampicillin, and ofloxacin medications, whereas it was susceptible to polymyxin-B, ertapenem, and nitrofurantoin (Appendix A and Appendix A).

### 2.2. Isolation and Characterization of Bacteriophages

BPs were successfully isolated and named BPABΦ1. The BPs were then quantified using DAOM. The BPABΦ1 plaques showed clear plaque with a white band (Figure 1A), typical for lytic (virulent) phage. Moreover, the plaques appeared round, with an average plaque diameter of 6.68 ± 0.94 mm.

Based on spot test analysis and confirmed by DLAO, we observed that BPABΦ1 showed a narrow host range, affecting only 27.90% of the total 43 bacterial hosts (Appendix A). Hence, the data suggested that phages obtained from MDR *A. baumannii* have narrow host-specificity, thereby making the BPs therapy an ideal candidate for personalized treatment.

To determine the proliferation rate, the latent period, and the burst size of BPABΦ1 per infected bacterial cell, the one-step growth curves study was performed on BPABΦ1. The analysis was performed in triplicates, and the obtained data were analyzed (Figure 1C). It was observed that the BPABΦ1 phage had a latent period of 25 min with a burst size of 102.12 ± 2.65 PFUs per infected host cell.

The phage suspension was loaded onto a copper grid and stained with 2% uranyl acetate, and the morphology was observed under the transmission electron microscope. The phage particles (Figure 1B, the arrows indicated) showed a hexagonal head about 85.93 ± 3.93 nm in diameter but without a neck and a tail, which categorized them under the *Corticoviridae* family [17].

Further, to determine the optimal pH and temperature conditions required for developing BPABΦ1-loaded formulations, thermal, pH, and UV stability study were performed. The PFU of BPABΦ1 titer was higher in the 6.8–7.4 pH and 4–40 °C temperature range. However, no active phages were found at pH 1.5 and 9.5 and a temperature above 70 °C (Figure 1D,E). Furthermore, it was observed that within 5 min of UV exposure, the complete titer of the phages was eradicated.

### 2.3. Formulation, Optimization, and Characterization of Bacteriophage Loaded Microparticle Laden Gel

In the compatibility study, no decrease in BPABΦ1 titer was observed when BPABΦ1 was exposed to different excipients selected for the formulation of bacteriophage-loaded microparticles.

The particle size, polydispersity index (PDI), zeta potential, and entrapment efficiency EE (%) of the optimized formulation (BPABΦ1-CHMPs) are shown in Table 1**.** The optimized microparticles were further incorporated into Sepineo^TM^ P 600 to convert the microparticles into a gel (BPABΦ1-CHMPs-Gel). Furthermore, BPABΦ1-CHMPs gel was found to be odorless, semi-transparent, homogenous, and easily spreadable with a viscosity of 2914.03 ± 51.17 centipoise.

#### 2.3.1. Surface Morphology

SEM was used to determine the surface morphology of the BPABΦ1-CHMPs. The morphology of the optimized BPABΦ1-CHMPs (Figure 2A) was found to be homogeneous and somewhat spherical. Moreover, in the case of BPABΦ1-CHMPs gel, the morphology looked like an interwoven network of irregular fibers (Figure 2B).

#### 2.3.2. In Vitro Bacteriophage Release Study

In vitro release profiles of free BPABΦ1, BPABΦ1-CHMPs, and BPABΦ1-CHMPs gel are depicted in Figure 2C. From the study, it was observed that 73.52 ± 1.68% BPABΦ1 was released from the solution within 30 min, whereas 98.34 ± 2.14% and 89.09 ± 2.85% BPABΦ1 were released from BPABΦ1-CHMPs and BPABΦ1-CHMPs-gel, respectively, after 12 h.

#### 2.3.3. In Vitro Antibacterial Studies

Spot tests were performed to assess the antibacterial efficacy of the formulations qualitatively. It was observed that the blank CHMPs showed very minute and unclear activity, while the blank gel exhibited no antimicrobial action. On the other hand, BPABΦ1-CHMPs and BPABΦ1-CHMPs gel both exhibited improved antibacterial activity against *A. baumannii*.

The quantitative antibacterial activity of formulation was performed, where MIC values of BPABΦ1, blank CHMPs, and BPABΦ1-CHMPs were found to be 5.6 × 10^6^ PFU/mL, 480 ± 3.46 µg/mL, 200.0 ± 2.66 µg/mL, respectively, after 24 h of incubation. Additionally, the MBCs values of BPABΦ1, blank CHMPs, and BPABΦ1-CHMPs formulations were found to be 1.9 × 10^8^ PFU/mL, 710 ± 2.03 µg/mL, and 250.0 ± 3.68 µg/mL, respectively (Table 2).

#### 2.3.4. Antibiofilm Assay

The biofilm formation over burn wounds leads to the failure of standard burn treatment. In this context, it was revealed that the treatment that can eradicate biofilm is efficient against burn wounds. From the study, it was observed that BPABΦ1 (88.69 ± 3.11%) and BPABΦ1-CHMPs (94.46 ± 2.19%) showed a significant (*p* < 0.05) biofilm eradication; however, only 21.57 ± 0.66 % of biofilm eradication was observed in case of blank CHMPs following the 24 h treatment, which was found to be significant, compared to BPABΦ1 and BPABΦ1-CHMPs (Table 2 and Figure 2F).

The visual microscopic examination of biofilm allows a better understanding of the spatial arrangement within the biofilms and their supporting surfaces. The SEM analysis found that the control biofilms (untreated Figure 2D) showed complex, multi-layered cell clusters embedded in a matrix of extracellular biological material. However, after applying the treatment with BPABΦ1-CHMPs, the area of the developed biofilm gets significantly reduced (Figure 2E) compared to the biofilm exposed with the control sample.

#### 2.3.5. Stability Study

The BPABΦ1 solution was found to be unstable (Figure 3), as it showed ~80% decreased titer within a month at a temperature of 4.0 ± 0.2 °C, with minimal antibacterial activity within 3 months. However, the entrapped bacteriophage titer in the cases of BPABΦ1-CHMPs and BPABΦ1-CHMPs gel (Figure 3) first declined (first month) by one-tenth titer, but after that, it remains stable for up to three months at 4.0 ± 0.2 °C. This stability could be due to the protective nature of trehalose and glycerin in the developed formulation.

### 2.4. Animal Study

#### 2.4.1. In Vivo Burn Wound Healing Assay Imaging by Ultrasound (USD) and Photoacoustic (PA) Imaging System

The evaluation of burn wound healing depends on how much area and depth are reduced following the treatment. From the study, it was observed that a higher percentage of wound closure (94.45 ± 2.52%) was observed with BPABΦ1-CHMPs gel (Group 3) (Figure 4A) as compared to the marketed formulations (Group 2: 79.21 ± 2.65%) on the 28th day. However, the control group exhibited only 26 ± 1.03% of wound contraction with increased pus formation, leading to increased wound area due to profuse infections.

A 3D-mode ultrasound (USD) was employed to collect the area (two-dimensional data) and depth-volume (three-dimensional data) of the wound. USD identified the wound location and evaluated the volume (mm^3^) of the wound on days 3, 14, and 21 by processing the image by multislice method from the upper surface to the underlying layer (Figure 5A), and quantitative wound volume changes were plotted, as shown in Figure 5B. On day 3, the wound volume was found to be 1482.81 ± 52.12, 986.71 ± 56.13, and 1632.29 ± 80.99 mm^3^ when treated with control (Group 1), 1% SSD (Group 2), and BPABΦ1-CHMPs gel (Group 3), respectively. It was further observed that the wound volume gets reduced by treatment with BPABΦ1-CHMPs-gel (85.12 ± 14.24 mm^3^) on day 21, although the smaller wound images were not visible in the 3D-USD images. However, it can be recognized in the 2D-USD and wound volume. Furthermore, the reduction in wound volume in 1% SSD (292.47 ± 39.54 mm^3^) was less than the prepared formulation, and the control group showed a significantly less decrease in wound volume (1195.42 ± 42.23 mm^3^) due to severe infection.

Photoacoustic imaging (PA-mode) was used to determine saturated oxygen percentage (sO_2_%). PA imaging (Figure 5C) is ideally suited to monitor local angiogenesis, perfusion, and oxygen saturation, which are the key parameters for wound healing. The day 3 oxygen saturation (sO_2_%) of control, 1% SSD, and BPABΦ1-CHMPs gel was found to be 16.91 ± 1.12%, 21.74 ± 2.00%, and 21.02 ± 2.29% (Figure 5D). sO_2_% gradually increased with increasing the day of treatment, and it was found higher in formulation group BPABΦ1-CHMPs gel (84.51 ± 2.24%) compared to the marketed formulation (64.16 ± 3.54%). However, in the control group, the sO_2_% increased (53.24 ± 2.72%) until day 14, which decreased (32.33 ± 1.14%) on day 21 due to serious infection.

#### 2.4.2. Histopathological Examination

The histological HE-stained images of sacrificed rat skin (Figure 4B) confirmed the faster and intact re-epithelialization of the BPABΦ1-CHMPs gel compared to the marketed formulation. On the contrary, untreated animals do not exhibit complete re-epithelialization even on day 28.

## 3. Discussion

Multidrug-resistant *A. baumannii* has emerged as one of the predominant organisms responsible for bacterial infections in burn wounds. Moreover, the infections in burn wounds facilitate the development of Biofilm around the wounded area, limiting the standard antibiotic treatment. In this context, *A. baumannii* were found resistant to more than one antibiotic, such as piperacillin/tazobactam, cefepime, co-trimoxazole, amikacin, ceftriaxone, gentamicin, cefalexin, ampicillin, and ofloxacin, which makes them an ideal antibiotic-resistant strain, hence showing multidrug resistance and exhibiting an incredible potential of forming biofilms that are difficult to eradicate. In this study, BPABΦ1 (BPs against MDR *A. baumannii*) was successfully isolated from *A. baumannii* (BHU/AB/39) multidrug-resistant bacteria and quantified. The host range of BPABΦ1 was found to be narrow, indicating its suitability for personalized treatment. However, some isolated phages (AGC01, Bϕ-R2096, ϕAB2, etc.) demonstrated broad-spectrum activity against *A. baumannii* bacteria while having no effect on other bacterial strains [18,19,20]. The morphology analysis of BPABΦ1 further confirms their categorization of *Corticoviridae* as they lack a neck and tail. Regarding the pH and thermal stability, the BPABΦ1 showed excellent stability between pH 6.8–7.4 and temperatures of 4–40 °C, which ensures that the phage will remain stable during its development into a suitable dosage form. Previous studies have documented better phage stability at pH 7–8 and temperatures of 21–45 °C [21]. Feng et al. (2003) studied the effects of temperature and pH on the survival of coliphages (MS2 and Qβ) in water and wastewater. The pH level of 6–8 and the temperature ranges of 5–35 °C showed the lowest inactivation rate for both phages [22]. Studies have proved pH < 4 and > 9 and temperatures > 40 °C are common limiting factors for phage activity [23,24].

Additionally, the weakly acidic and weakly basic pH facilitates the fabrication of topical dosage form, as the oral route might deteriorate the BPABΦ1 along the gastrointestinal tract. Moreover, the one-step growth curve revealed that the BPABΦ1 exhibits an extremely rapid lytic activity, with a short latent time of only 25 min and a relatively larger average burst size.

Further, to deliver the BPABΦ1 effectively into the affected site, the BPABΦ1 was loaded in chitosan-based microparticles (BPABΦ1-CHMPs). The developed formulation corresponded to the desired variable: particle size within 1–5 μm, homogenous PDI, increased entrapment, and positive zeta potential. It was revealed from various studies that formulation of particle size 1–5 μm can easily permeate the follicular channels and target follicular disorders such as bacterial infections without disturbing the systemic effects of the drug, thereby mediating an improved local impact of the BPABΦ1-CHMPs over the burn wounded region when applied topically. Also, the positive zeta potential is implied due to positively charged chitosan, which helps to target the bacterial negative charge cell wall. It was also demonstrated that chitosan showed an additive antibacterial effect. The microparticles were further converted into gel to improve their thermodynamic and pharmacokinetic parameters. The developed BPABΦ1 loaded microparticle-laden gel was also stable with good organoleptic properties ideal for their use as a wound healing formulation. The in vitro release study demonstrated the faster release of free BPABΦ1, which could be assigned to the absence of any controlling barrier except the dialysis membrane. The microparticles depicted initial burst release up to 1.5–2 h, which may be attributed to the release of BPABΦ1 adhered to the microparticle’s surface and the faster diffusion of BPABΦ1 present just beneath the surface of the microparticles. Later on, sustained release of BPABΦ1-CHMPs may be attributed to the slower diffusion of the BPABΦ1 from the core of microparticles. BPABΦ1 is distributed into the microparticle polymeric matrix, and, hence, the BPABΦ1 may take a longer time to reach the microparticle surface. Additionally, the incorporation of the BPABΦ1-CHMPs into the gel further creates an additional barrier for the diffusion of the BPABΦ1 from microparticles to gel and then finally to the release medium. Hence, loading of the microparticles in the gel further enhances the sustained release of the BPABΦ1 from the microparticles. Likewise, the ϕAB bacteriophages were encapsulated with chitosan and integrated into a hydrogel matrix, resulting in a sustained release of the bacteriophages for 6 h [10]. Similarly, microencapsulated BPs in poly (lactic-co-glycolic acid) were developed using a double emulsification approach by Jamaledin, R. et al., 2023. A sustained release was observed with bacteriophage-loaded MPs formulations, with a rapid initial release followed by a slower release over time [25].

The in vitro study observed a higher antibacterial effect on BPABΦ1 and BPABΦ1-CHMPs gel, a lessened antibacterial effect in blank CHMPs, and no antibacterial effect in the blank gel.

Additionally, applying BPABΦ1-CHMPs gel over the wounded area improves the BPABΦ1 titer on the wounded site, limiting its leakage and enhancing poor in vivo retention and inactivation by the immune cells present in the neighboring province of the wounded region. BPABΦ1-CHMPs gel provided a more sustained release profile than BPABΦ1-CHMPs, prolonging their contact time and increasing the antibacterial activity. The SEM results further stated that the BPABΦ1-CHMPs gel displayed an interwoven network that aids in encapsulating the BPs more firmly, and this phenomenon could be assigned to the most sustained release observed in the case of BPABΦ1 CHMPs-gel.

Over and above that, the data from in vivo wound healing studies using USG/PA imaging were supported by the findings of histopathological examinations. Ultrasound images clearly showed the more severe damage to the skin layer on day 3, and as the burn wound recovered, the burn depth started to decrease gradually (day 14). Since post-burn day 21, burn depth had maximum turned down to a low level on day 21 by using bacteriophage formulation. Proving that BPABΦ1-CHMPs-gel could facilitate complete wound closure by day 28. Researchers reported a similar result; for example, lytic phases ϕAB140 and ϕAB150 chitosan formulation demonstrated considerable wound size reduction caused by MDR bacterial infection [10]. Although the control group showed approximately 26% of wound contraction, pus and bad odor were observed, denoting the inability to achieve complete bacterial clearance and healing of the wound.

Additionally, the percentage of sO_2_% exhibited a gradual increase as the duration of treatment increased. Moreover, it was observed that the formulation groups had higher levels of sO_2_%.

On the contrary, in the control group, sO_2_% was less in the initial days (day 3), which later increased and decreased. Such a phenomenon was caused due to bacterial biofilm infection, which could not heal the wound. The most interesting observation in PA images was found to be more hypoxia in the central area of the burn as compared to the surrounding area in the first week, which could be correlated with dermis damage and lack of blood perfusion. Similar conclusions from studies were made by Yash Mantri et al. in 2021, who stated that scar tissue production, angiogenesis, and wound size could all be measured using PA-US imaging. Researchers are now able to see angiogenesis around the healing wound because of the addition of PA imaging [26]. The data from in vivo wound healing studies were supported by the findings of histopathological examinations. BPABΦ1-CHMPs-gel showed epidermal healing, collagen formation, and a good amount of granulation around the wounded area compared to the control group. This could be attributed to the specificity of BPABΦ1 against antibiotic-resistant strains, which effectively ceases the growth of invading bacteria at the wounded region. BPABΦ1-CHMPs gel provides an effective therapy that readily delivers the wounded animals without delay, making phage therapy potentially more effective than marketed formulations. Further, the stability of BPABΦ1 in the formulation was maintained for more than 4 months with effective titer. However, previous research showed that lyophilized phage stored using trehalose remained viable and lyrically active for up to 126 days following resuspension in saline, although at a reduced titer [27].

We did not observe any sign of adverse effects or toxicity during and after treatment with the developed formulation. That is an indication of good formulation.

## 4. Materials and Methods

### 4.1. Chemical Reagent Used

Luria Bertini (LB), TMG buffer (Tris hydrochloride, magnesium sulfate, gelatin buffer), Muller Hinton Agar (MHA), Agar powder (Bacteriological grade), all antibiotic discs, and media were supplied by Himedia, Maharashtra, India. Trehalose dihydrate, chitosan oligosaccharide (extra pure, 90% degree of deacetylation), and sodium tripolyphosphate (STPP) were purchased from Sisco Research Laboratories, Mumbai, India. Glycerol (molecular biology grade, extra pure), DiD (DiIC18(5); 1,1′-dioctadecyl-3,3,3′,3′- tetramethylindodicarbocyanine, 4-chlorobenzenesulfonate salt) were purchased from, Thermo Fisher, Waltham, MA, USA. All other chemicals and reagents used in this study were molecular biology grade, bacteriological, and lab grade.

### 4.2. Bacterial Strain Collection and Antibiotic Susceptibility Test

*A. baumannii* (BHU/AB/39) multidrug-resistant bacteria were isolated from the pus of an injured patient and provided by Prof. Gopal Nath, Department of Medical Microbiology, Institute of Medical Sciences (IMS), Banaras Hindu University, Varanasi (India). Further, *A. baumannii* was identified by biochemical and molecular methods (16S rDNA and rec A genes analyses) and published by Patel et al. (2021) [3]. Antibiotic sensitivity was performed by the Kirby–Bauer disc diffusion method using 11 antibiotic discs (Appendix A) viz. piperacillin/tazobactam (100/10 µg), cefepime (30 µg), co-trimoxazole (25 µg), amikacin (30 µg), ceftriaxone (30 µg), polymyxin B (300 units), gentamicin (10 µg), cefalexin (30 µg), ampicillin (10 µg), ertapenem (10 µg), nitrofurantoin (300 µg), and ofloxacin (5 µg). The bacterial suspension (0.5 McFarland turbidity standard) was lawned on Mueller–Hinton agar petri plates, followed by an antibiotic disc placed and incubated at 37 °C for 18 h. The inhibition halos diameter was measured in millimeters (mm), and isolates were classified as susceptive or resistant by comparing the data with the established standard charts [28].

### 4.3. Isolation and Characterization of Bacteriophage

The BPs were isolated by using the traditional enrichment method [29]. The water samples (2 mL, collected from the Ganges River) were centrifuged at 12,000× *g* for 10 min to remove debris and coarse material. The supernatant was then serially passed through membrane filters with pore sizes of 0.45 μm and 0.22 μm (Millex-GV Filter, Merck, Darmstadt, Germany) [30]. The processed water samples were then dropped on lawned *A. baumannii* bacterial Mueller–Hinton agar (MHA) petri dishes and incubated overnight at 37 °C. After incubation, the petri dishes were examined for the presence of lytic bacteriophage plaques and harvested with TMG buffer. The culturing and harvesting procedures were repeated until a total bacterial clearance and a higher BP (BPABΦ1) titer were obtained. The BPs were then purified by a single sequential plaque isolation method using a double-layer agar overlay technique (DLAO) [31].

#### 4.3.1. Quantification

The quantification of BPABΦ1 was performed using the double-layer agar overlay (DLAO) method and measured in terms of the phage titer value (PFU/mL). Briefly, 1 mL of each BPABΦ1 sample and bacterial host (200 µL of 1.0 optical density (OD)) was added to the separate soft agar (0.75%) in molten condition. The phage-host suspension in soft agar was overlaid on solidified bottom agar in 90 mm petri plates. Plates were then gently swirled, dried at room temperature for 10 min, and incubated overnight at 37 °C. The next day, the BPABΦ1 titer was enumerated by the following equation [32];
$$Bacteriophage\;titer\;\left(PFU/\mathrm{mL}\right)=\;\frac{numberof\;plaques\;per\;\mathrm{mL}}{Dilution\;factor}$$

After getting plaques from the DLAO assay, the plaque morphology was characterized based on the plaque diameter, a halo zone around the plaques, and bacterial clearance [33,34].

#### 4.3.2. Host Range Determination

The host range of the isolated BPABΦ1 was identified by spot test. Briefly, 20 µL of BPABΦ1 were dropped in 43 *A. baumannii* bacterial clinical isolates (Appendix A) on lawned petri plates and inspected for lysis zones after overnight incubation at 37 °C [35].

#### 4.3.3. Burst Size Identification

A one-step growth curve was performed to monitor the growth of BPABΦ1 in a respective bacterial host. This study helps to identify the burst size of BPABΦ1. The one-step growth curve was performed by the traditionally used method, as reported by Ellis E.L. and Delbruck M. [36,37], with slight modifications. Briefly, freshly grown *A. baumannii* (2.4 × 10^6^ CFU/mL) were centrifuged (10 mL, 6000 g for 10 min) and mixed with 10 mL of respective phage (BPABΦ1) while maintaining the multiplicity of infection (MOI, 0.01) and incubated for 10 min (permit phage adsorption at the bacterial surface) at 37 °C with aeration (150 rpm). The bacteria and BPABΦ1 were collected in 2 sample sets after different time points (5, 10, 15, 20, 25, 30, 35, 40, 50, 60, 55, and 60 min). The first sample set was instantly diluted and plated for BPs titration by the DLAO. A second sample set was processed with chloroform (1% *v*/*v*) to release intracellular BPs before phage titration for assessing the eclipse time. The time series data were plotted using GraphPad prism, and the average burst size per infected host and average latent period was determined from the sigmoidal curves.

#### 4.3.4. Morphological Evaluation

Transmission electron microscopy (TEM) was used to identify BPABΦ1 according to their size and form. BPABΦ1 (~13.5 mL, ~10^9^–10^10^ pfu/mL) were ultracentrifuged (XPN 100 Ultracentrifuge, Beckman Coulter Inc., California, USA) at 30,000× *g* for 60 min. The pellets were collected and washed three times with 0.1 M ammonium acetate (13.6 mL and pH 7.0). The pellets were then re-dispersed in 200 µL of ammonium acetate, and 5 µL were loaded in a carbon-coated TEM grid (#400), which were then negatively stained using phosphotungstic acid (5 µL). The morphology of the BPABΦ1 was examined using Cryo-TEM (TALOS S, Thermo Scientific at SAIF-AIIMS Delhi) [38].

#### 4.3.5. Thermal, pH, and UV Stability Test

The stability of the BPABΦ1 (9 × 10^7^ PFU/mL) was performed under different pH conditions (1.5, 3.5, 6.8, 7.4, and 8.5 at 37 °C for 24 h), different temperatures (4, 25, 37, 45, 60, and 70 °C at pH 7.4 for 60 min) and in the presence of UV light (0, 5, 10, 15, and 20 min). The BPABΦ1 samples were incubated under the abovementioned conditions, followed by ice-cooling at 4 °C. The titer obtained was then assessed using the DLAO method [39].

### 4.4. Formulation, Optimization, and Characterization of Bacteriophage Loaded Microparticle Laden Gel

Before proceeding further with formulation development, the BP’s compatibility with excipients (chitosan, sodium tripolyphosphate, trehalose, glycerol) was tested by maintaining a 1:1 ratio of each excipient with BPs in LB broth for 24 h at 37 °C. Finally, the BP titer was determined using the DLAO method [40].

#### 4.4.1. Bacteriophage Microparticle Preparation

Chitosan microparticles (CHMPs) of BPABΦ1 (BPABΦ1-CHMPs) were formulated by using the ion gelation method (Table 3). Briefly, the polymeric solution was prepared by dissolving chitosan (600 mg) and D (+)-Trehalose dihydrate (TD, 0.5% *w*/*v*) in PBS (50 mL) and passed through a 0.45 µ sterile syringe filter. Under sterile conditions with continuous magnetic stirring (200 rpm) at room temperature (25 °C), the aqueous solution of STPP (14 mg) was prepared and added dropwise using 28 gauge syringes into the polymeric solution, resulting in the fabrication of blank CHMPs. To develop the BPABΦ1-loaded CHMPs, 1.0 mL of the BPABΦ1 solution was added separately to the polymeric solution; similarly, aqueous STPP solution was added dropwise with continuous stirring. The developed BPABΦ1-CHMPs were centrifuged at 6000 rpm for 10 min, and pellets were washed twice with PBS (3 mL) and lyophilized [10,41]. Even though the formulation technique was straightforward, keeping a sterilized environment, supplying ambient temperature, and maintaining a neutral pH are required since these impact the survivability of phages. Furthermore, endotoxins and other contaminants in phage preparations should be avoided.

#### 4.4.2. BPABΦ1-CHMPs Laden Gel

BPABΦ1-CHMPs laden gel was prepared under aseptic conditions in laminar flow. Briefly, filtered SEPINEO™ P 600 (2.5% *v*/*v* of total formulation 10 mL) was mixed with 5 mL of BPABΦ1-CHMPs (1.0 g of the lyophilized microparticles), and glycerol (0.5%) using continuous stirring at 200 rpm for 30 min to form a uniform gel. Later, the prepared gel was kept overnight for proper gel swelling, which further resulted in BPABΦ1 loaded microparticles laden gel (BPABΦ1-CHMPs gel) [42]. Similarly, the blank gel was prepared using the same process but without incorporating of BPABΦ1-CHMPs.

#### 4.4.3. Analysis of Particle Size, Polydispersity Index, and Zeta Potential

Particle size, polydispersity index (PDI), and zeta potential of prepared BPABΦ1-CHMPs were determined using zeta sizer (Malvern Zetasizer Nanoseries, S90) by employing dynamic light scattering and electrophoretic mobility method. Before analysis, the samples were dispersed in Millipore water with a dilution factor 10 [43].

#### 4.4.4. Determination of Entrapment Efficiency

Entrapment efficiency was determined by the indirect method. The BPABΦ1-CHMPs dispersion was centrifuged at 15,000 rpm at 4 °C for 20 min. The supernatant was collected and serially diluted [44]. The amount of BPABΦ1 present in the supernatant was determined by the DLAO method. The following formula was used to determine the entrapment efficiency (EE%);
$$EE\%=\frac{\mathrm{Total}\;\mathrm{amount}\;\mathrm{of}\;\mathrm{incorporated}\;\mathrm{BP}-\mathrm{free}\;\mathrm{BP}}{\mathrm{Total}\;\mathrm{amount}\;\mathrm{of}\;\mathrm{incorporated}\;\mathrm{BP}}\times 100$$

#### 4.4.5. Characterization of BPABΦ1-CHMPs-Gel

A visual inspection determined the organoleptic appearance (clarity, odor) and gel homogeneity. The viscosity of the prepared BPABΦ1-CHMPs-gel was measured using a Brookfield viscometer (DVE, LV Spindle no. 61). Briefly, the viscometer spindle was dipped in a beaker containing 20 g of gel and rotated at 50 rpm under room temperature, and viscosity of the samples was measured in centipoise [45]. Additionally, the prepared gel was screened for spreadability. The gel (0.5 g) was transferred on a 2 cm circle marked glass palate. The spreadability of the gel was then examined by placing a second glass plate on the top, with 500 g of weight, for five minutes. The circle’s increasing diameter was measured after the gel was spread [46]. Every measurement was made three times with a fresh sample each time.

#### 4.4.6. Surface Morphology

Surface morphology of the prepared BPABΦ1-CHMPs and BPABΦ1-CHMPs gel was examined using scanning electron microscopy (SEM; EVO-SEM, MA15/18, CARL ZEISS MICROSCOPY LTD, Oberkochen, Germany). Briefly, a drop of diluted BPABΦ1-CHMPs was cast on a coverslip and left to dry overnight under a vacuum, followed by SEM imaging. In addition, BPABΦ1-CHMPs gel and blank gel were lyophilized and subjected to SEM imaging (BT SEM; JEOL 6000, Oberkochen, Germany) [47].

#### 4.4.7. In Vitro Release Study

In vitro release of BPABΦ1 from BPABΦ1-CHMPs, and BPABΦ1-CHMPs-laden gel was carried out using the dialysis bag method [47]. Briefly, BPABΦ1-CHMPs and BPABΦ1-CHMPs gel samples were filled in a pre-activated dialysis bag (MWCO 12 kDa) and suspended in 100 mL of phosphate-buffered saline (PBS) (pH 7.4) in a glass beaker, maintained at 37 °C, and continuous stirring at 50 rpm. Samples (1 mL) were periodically withdrawn and replaced with an equal volume of fresh medium. The amount of BPABΦ1 released from CHMPs and CHMPs gel was analyzed using the DLAO. The in vitro release studies were performed in triplicate for each BPABΦ1 sample.

### 4.5. In Vitro Antibacterial Studies

The qualitative antibacterial activity of blank CHMPs, BPABΦ1-CHMPs, blank gel, and BPABΦ1-CHMP gel was determined by spot test against *A. baumannii.* In brief, 20 µL from different samples were dropped on a bacterial lawned (0.5 McFarland) MHA plate and incubated overnight at 37 °C. After incubation, the activity of the samples was checked by observing and measuring the clear zone.

Further, for quantitative antibacterial studies, minimum inhibitory concentration (MIC) and minimum bactericidal concentration (MBC) of microparticles were carried out by following the CLSI guidelines on planktonic cultures by the micro-broth dilution method on *A. baumannii* host. In brief, *A. baumannii* were grown in LB broth (1.5 × 10^8^ CFU/mL), and 100 µL of bacterial suspension was filled into 96 well plate. Then serially diluted 100 µL of BPABΦ1 and BPABΦ1-CHMPs (ranging from 1000–10 µg/mL) were added to the separate well. Corresponding bacterial controls (LB-grown bacteria), phage controls (phage stock solution), and media controls (LB broth) were maintained. The microtiter plates were incubated at 35 °C for 24 h with gentle shaking at 20 rpm. The lowest concentration of BPABΦ1 at which no turbidity was seen was regarded as the MIC [47].

MBC was also assessed using the CLSI protocol. A sample (100 µL) was obtained from wells of microplates where no apparent growth was seen after 24 h of culture at 37 °C and inoculated onto the surface of MHA plates. The resultant sample was maintained at 37 °C overnight, with MBC indicating the minimum concentration of the substance, where no colonies were formed. The lack of growth on the MHA plate suggested that the concentration of BPABΦ1 was under the detection limit for this method: 10 CFU/mL [48].

#### 4.5.1. Antibiofilm Assay

The ability of BPABΦ1 to eradicate mature biofilms is much more important than the ability to inhibit its formation. A previously established quantitative microtiter plate assay (crystal violet) was used to assess the antibiofilm efficacy of BPABΦ1, blank CHMPs, and BPABΦ1-CHMPs [49,50]. In brief, 180 µL of LB growth media was added to 20 µL of bacteria (*A. baumannii* (0.5 OD_600nm_)) and incubated in a sterile 96-well plate at 37 °C for 48 h. Further, planktonic bacteria were aspirated, and samples (200 µL, 2× dilution of MIC) were added to the respective well, followed by incubation for 24 h. The sample of the microplate wells was aspirated after incubation and thoroughly rinsed with PBS (pH 7.4) to eliminate the free-floating non-adherent bacteria. The microplate wells were then air-dried for 60 min. Following drying, the adhering “sessile” bacteria in the wells were fixed with 2% *w*/*v* sodium acetate before being saturated with 0.1% *w*/*v* crystal violet dye and placed in a darkened room for 30 min. The wells were then carefully cleansed with deionized water to eliminate any remaining color. After drying the plate completely, 200 µL of ethyl alcohol (95%, *v*/*v*) was introduced to each well, and absorbance was measured at 620 nm (Multiscan FC microplate reader, Thermo Fisher Scientific, India). Wells containing only growth media were inoculated with tested bacterial isolate and were used as a control. The mean value of the three measurements was reported. The percentage inhibition of biofilm formation was calculated using the following equation:$$Inhibition\;\%=\frac{Optical\;density\;of\;control-Optical\;density\;of\;treatment\;}{Optical\;density\;of\;control}\times 100$$

#### 4.5.2. Scanning Electron Microscopy of BIOFILM

An overnight culture of *A. baumannii* bacteria was dispensed into a 12-well culture plate with a sterile round coverslip and incubated for 48 h. After incubation, the formed biofilm was treated with BPABΦ1, blank CHMPs, and BPABΦ1-CHMPs in the required concentration and then incubated for another 24 h. The SEM confirmed the eradication of biofilm (Carl Zeiss Microscopy Ltd. Oberkochen, Germany).

#### 4.5.3. Stability Studies

BPABΦ1 solution, BPABΦ1-CHMPs, and BPABΦ1-CHMPs gel were sealed and stored at 4 °C for 7 months. After each month, the required sample was placed in a dialysis bag, and a release study was performed in 50 mL of sterile PBS (pH 7.4) for 24 h. The livability and lytic activity against host bacteria were checked using the DLAO method [51].

### 4.6. Animal Study

All animal experiments were approved by IAEC, Dept. of Pharmaceutical Engineering and Technology, Indian Institute of Technology (BHU), Varanasi, U.P., India. The study used 15 female/male Wistar rats (7–9 weeks, 200–250 g weight).

#### 4.6.1. In Vivo Burn Wound Healing Assay Imaging by Ultrasound and Photoacoustic (PA) Imaging System

For developing the burn wound, animals were first anesthetized with an intraperitoneal injection of ketamine and xylazine (80 mg/kg and 20 mg/kg). A cylindrical stainless-steel rod (1.5 cm diameter) was heated to 100 °C in boiling water and placed in the dorsal region of the rat for 20 s. After 6 h of wound creation, 100 µL of *A. baumannii* bacteria 1 OD was swabbed on the wound area, and 20 µL injected subcutaneously for two successive days to induce bacterial infection and robust biofilm formation in the wound. After the development of burn-induced wounds, the animals were randomly divided into three groups (*n* = 5)—Group 1: control group (no treatment), Group 2: treatment with marketed formulation, Silvadene^®^ cream (silver sulfadiazine (SSD 1.0%), Group 3: treatment with BPABΦ1-CHMPs gel. Phage formulation (0.5 g) was applied twice daily, and the wound healing potency of BPABΦ1-CHMPs gel was evaluated by measuring the wounded area following the different treatments by a scale on different days until complete re-epithelialization. The wound healing was quantified using a scale:(1)$$\%\;\mathrm{wound}\;\mathrm{retraction}\;\mathrm{on}\;\mathrm{day}\;X=\frac{\left(\mathrm{wound}\;\mathrm{area}\;\mathrm{on}\;\mathrm{day}\;\mathrm{zero}\;-\;\mathrm{wound}\;\mathrm{area}\;\mathrm{on}\;\mathrm{day}\;X\right)}{\left(\mathrm{wound}\;\mathrm{area}\;\mathrm{on}\;\mathrm{day}\;\mathrm{zero}\right)}$$

Simultaneously, the wound volume and oxygen saturation were assessed by ultrasound/photoacoustic (USG/PA) imaging system (VisualSonics, Vevo F2 LAZR-X PA scanner, UHF 48 transducer, FUJIFILM VisualSonics Inc., Toronto, Canada) [52,53]. Additionally, angiogenesis, wound volume, and saturated oxygen percentage were also validated using ultrasound and photoacoustic imaging. The scanning of the wound was performed in B-mode (ultrasound imaging) and oxy-hemo mode at 750/850 nm laser for photoacoustic imaging. The following imaging parameters were set for photoacoustic imaging: PA gain = 37 dB, step size 0.33 mm, power 100%, and sensitivity was kept high. During imaging, rats were anesthetized by using 3% isoflurane (induction dose) and 1.5–2% maintenance dose. The animals were then laid prone position on an operating table maintained at 37 °C to perform the ultrasound and photoacoustic imaging. All the images were processed by using Vevo LAB software (FUJIFILM VisualSonics, Toronto, ON, Canada). Moreover, angiogenesis, wound volume, and saturated oxygen percentage were validated using ultrasound and photoacoustic imaging. The time required for re-epithelialization was calculated as the number of days needed for wound healing to the number of days required for the eschar to come off without leaving any raw wounds. The time required for complete re-epithelialization of wound was considered as endpoint of the experiment.

#### 4.6.2. Histopathological Examination

After the animal undergoes complete re-epithelization, the wound skin and highly perfused organs are removed and fixed in formalin before being processed and encased in paraffin. Sections were cut, where 5 µm sections were stained with hematoxylin and eosin. The optical microscope was used to examine the re-epithelization of wound skin and organ damage. Epithelization, inflammatory cell infiltration, fibroblast proliferation neovascularization, and collagen deposition were also observed under a light microscope (Olympus BX51; Olympus, Tokyo, Japan; magnification: ×100) [54].

### 4.7. Statistical Analysis

Data for the in vitro and in vivo experiments were presented as the mean ± SD (*n* = 3). GraphPad Prism 5.0 was used for statistical calculation. One-way ANOVA was used to analyze the statistical significance among the groups. The following statistically significant levels were considered as non-significant (ns) (*p* ≥ 0.05) and significant: * (*p* < 0.05), ** (*p* < 0.01), *** (*p* < 0.001).

## 5. Conclusions

Biofilm-mediated wound infections caused by MDR *A. baumannii* trains are sometimes challenging to treat with clinically available antibiotics aggravating the disease severity. Phage therapy has evolved as an effective treatment strategy for such cases; however, maintaining the bacteriolytic activity in the formulation is a great challenge. In this study, we have developed the chitosan microparticles by using a direct ionic gelation method, which is a very simple method widely reported for the preparation of the chitosan microparticles. The microparticles were formed by crosslinking and ionic interactions between the chitosan-free amine group and phosphate group of the sodium TPP. Further, loading of the developed microparticles into the “SEPINEO™ P 600” was also a very simple method. SEPINEO™ P 600 is a readily available vehicle, which, upon gentle stirring after addition of water, forms gels. Further, CHMPs gel was developed using BPABΦ1 to treat biofilm-mediated burn wound infections caused by MDR *A. baumannii*. The BPs were isolated and formulated as a gel for topical application with the desired quality attributes. The developed formulation was able to sustain the release of the BPs with increased wound closure and re-epithelization of the skin.

Interestingly, the entrapped phages’ biological activity was maintained for more than 4 months. Data suggest phage therapy’s superiority in treating biofilm-mediated wound infections due to MDR *A. baumannii*. Further efforts in the formulation designs are needed to improve the biological activity for even years to develop this therapy as an alternative to antibiotics.

## Figures and Tables

**Figure 1 pharmaceuticals-16-00942-f001:**
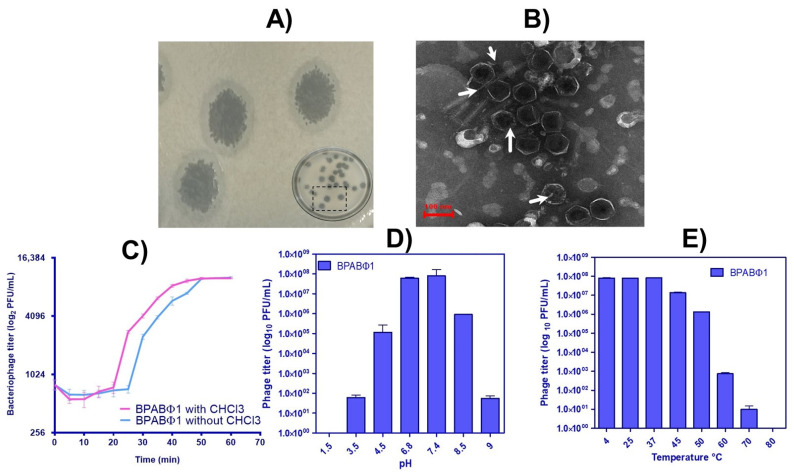
Characterization of bacteriophage isolates: (**A**) plaque assay of BPABΦ1, (**B**) morphological identification of BPABΦ1 by TEM analysis, (**C**) one-step growth curve of BPABΦ1, and (**D**,**E**) graphical representation of pH and temperature stability of BPABΦ1, respectively.

**Figure 2 pharmaceuticals-16-00942-f002:**
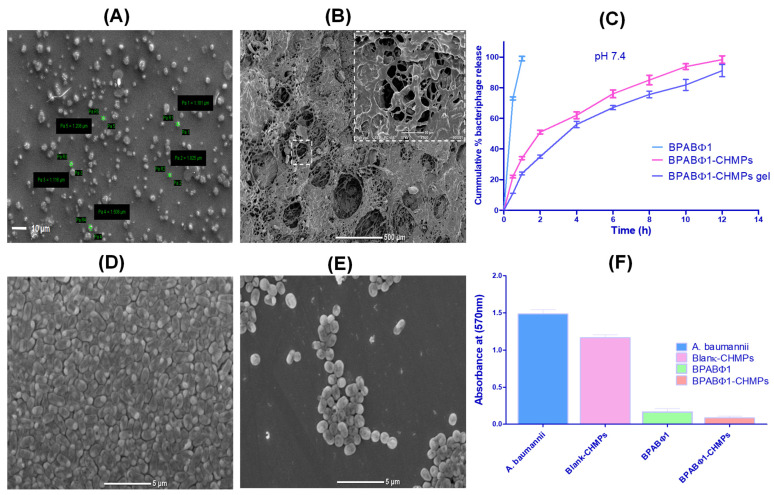
Characterization of formulation: (**A**) SEM image of BPABΦ1-CHMPs, and (**B**) SEM image of BPABΦ1-CHMPs *laden* gel. (**C**) Graphical representation of in vitro BPABΦ1 release study from BPABΦ1-CHMPs and BPABΦ1-CHMPs-laden gel. (**D**) *A. baumannii* biofilm eradication study of control. (**E**) *A. baumannii* biofilm eradication study with BPABΦ1-CHMPs. (**F**) Graphical representation of the anti-biofilm study.

**Figure 3 pharmaceuticals-16-00942-f003:**
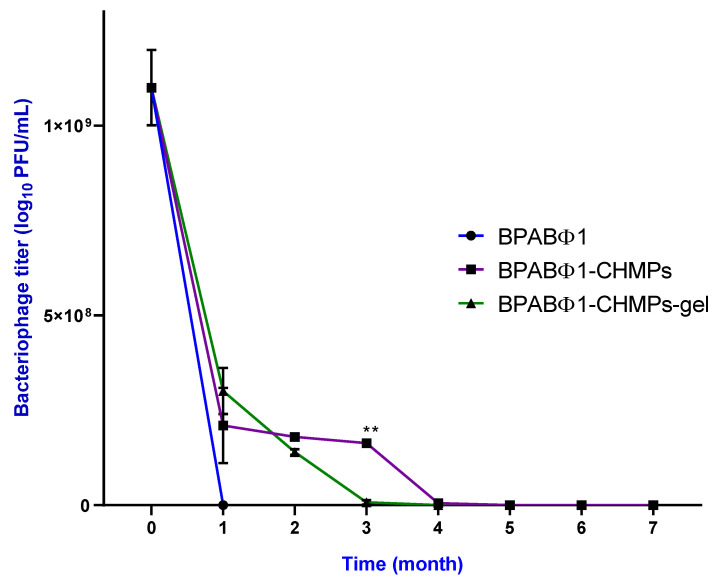
Stability study of isolated bacteriophage BPABΦ1, BPABΦ1-CHMPs, and BPABΦ1-CHMPs gel. The samples were stored at 4 °C for up to 7 months and were taken out every month to check the bacteriophage titer in the respective sample. The graph represents the bacteriophage titer on the X axis while the time in Months on the Y axis. Error bar represents standard deviation (*n* = 3), where ** significant difference at *p* < 0.01.

**Figure 4 pharmaceuticals-16-00942-f004:**
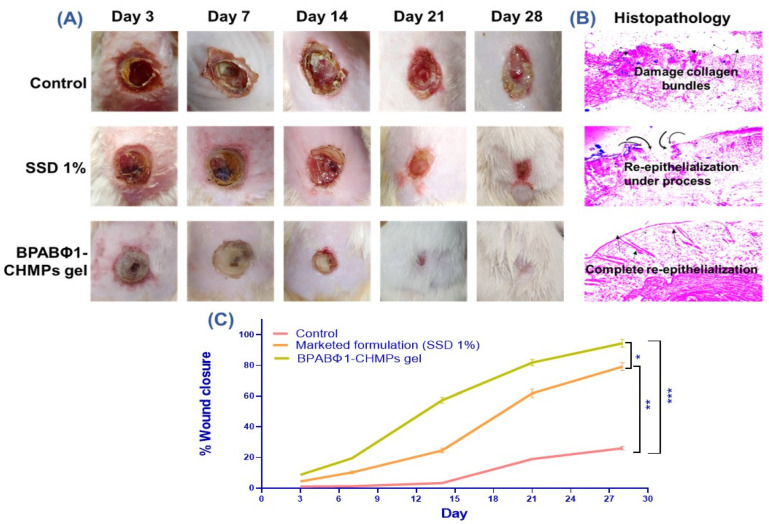
In vivo wound healing study on rat model where (**A**) shows wound status on days 3, 7, 14, 21, and 28; (**B**) histopathology examination of the skin of respected group; and (**C**) graphical data representation of % wound closure. Error bar represents standard deviation (*n* = 5), where * significant difference at *p* < 0.05, ** significant difference at *p* < 0.01 and *** significant difference at *p* < 0.001.

**Figure 5 pharmaceuticals-16-00942-f005:**
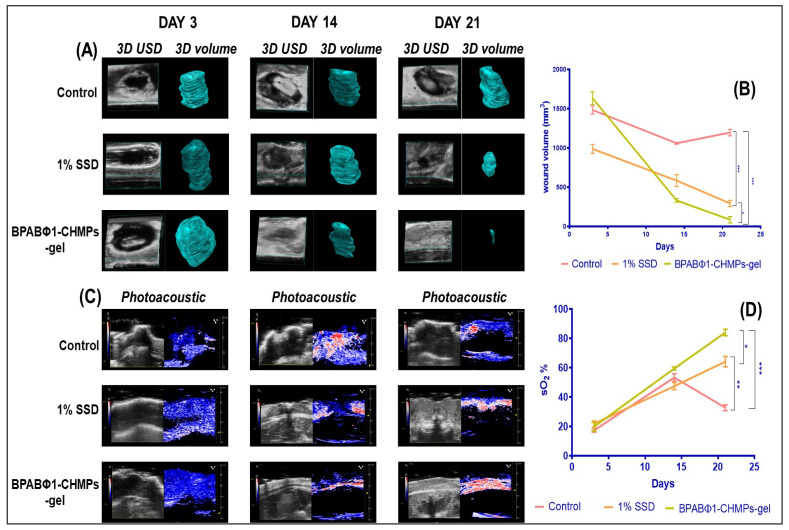
Ultrasound imaging of wound, where (**A**) 3D ultrasound imaging of the control group, 1% SSD (marketed formulation), and treatment with BPABΦ-CHMPs gel; (**B**) line graph indicating the change in wound 3D volume on the Y axis and time on X axis; (**C**) photoacoustic imaging of wound with (**D**) graphical representation of changes in sO_2_ % of different groups on day 3, 14, and 21. Error bar represents standard deviation (*n* = 5), where * significant difference at *p* < 0.05, ** significant difference at *p* < 0.01 and *** significant difference at *p* < 0.001.

**Table 1 pharmaceuticals-16-00942-t001:** Particle size, polydispersity, zeta potential, and entrapment efficiency of different formulations.

Group	Particle Size (µm)	PDI	Zeta Potential (mV)	% EE
Blank CHMPs	1.041 ± 0.244	0.212 ± 0.026	24.6 ± 0.18	-
BPABΦ1-CHMPs	1.291 ± 0.535	0.305 ± 0.011	36.41 ± 0.64	91.30 ± 1.94%

Data has been shown as Mean ± SD *n* = 3 *n* = 3; SD: Standard deviation.

**Table 2 pharmaceuticals-16-00942-t002:** Antibacterial efficacy studies of bacteriophage formulations.

Group	MIC	MBC	Biofilm Eradication (%)
BPABΦ1	5.6 × 10^6^ PFU/mL	1.9 × 10^8^ PFU/mL	88.69 ± 3.11
Blank CHMP	480 ± 3.46 µg/mL	710 ± 2.03 µg/mL	21.57 ± 0.66
BPABΦ1-CHMPs	200.0 ± 2.66 µg/mL	250.0 ± 3.68 µg/mL	94.46 ± 2.19

**Table 3 pharmaceuticals-16-00942-t003:** Composition of blank CHMPs and BPABΦ1-CHMPs.

Group	Chitosan (mg)	BPABΦ1 (PFU/mL)	Trehalose % (*w*/*v*)	STPP (mg)
Blank CHMPs	600	-	0.5	14.0
BPABΦ1-CHMPs	600	8.2 × 10^8^	0.5	14.0

## Data Availability

Data are contained within the article and the Appendix A.

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
