# Peer review of "A Bacteriophage Microgel Effectively Treats the Multidrug-Resistant Acinetobacter baumannii Bacterial Infections in Burn Wounds"

_pharmaceuticals, 2023, doi:10.3390/ph16070942_

Round 1

Reviewer 1 Report

Abstract:

1.     Corticoviridae should be italic.

Introduction

2.     The authors should add more details about the previous studies on molecules (or phages) loaded chitosan microparticles.

Methods

3.     The whole genome of the phage should be sequenced.

4.     EOP should be performed.

5.     Line 135: add India.

6.     The author should briefly describe about DLAO method.

Results

7.     Line 390: Phage should start with p.

8.     Fig 1(A) The enlarged figure is not corresponding with the plate. The author should check the ratio of the enlarged figure.

9.     Fig 3(c) and Fig 4(b,d) The author should perform the statistical analysis.

10.  To make it easier to understand, the author should change the bar graph to the line graph. A statistical analysis should be performed.

Conclusion

11.  The author performed the experiment against only A. baumannii so the author should change the word from MDR bacteria to MDR A. baumannii.

12.  According to the stability results, the author should rewrite the stability of the entrapped phage (Line 593-594).

Author Response

We sincerely thank the reviewer for his/her time and effort in providing valuable comments and suggestions. Corresponding changes have been made in red text in the revised version of the manuscript with track change mode, which is listed below point-to-point in bold text. The original comments are copied in italics text.

Comment 1: Abstract: Corticoviridae should be italic.

Response: As suggested by the reviewer, we have rectified Corticoviridae in the revised manuscript. Please see the line no. 16. 

Comment 2: Introduction: The authors should add more details about the previous studies on molecules (or phages) loaded with chitosan microparticles.

Response: As suggested by the reviewer, we have added more details about the previous studies on molecules (or phages) loaded with chitosan microparticles in the revised manuscript. Please see the line no. 67-92. The following research is included.

 “In a study, Ilomuanya et al. (2022) reported the formulation of the Acinetobacter baumannii phage cocktail loaded with chitosan microparticles for treating MDR wound infection. They have isolated the bacteriophages, namely, ɸAB140 and ɸAB150 against Acinetobacter baumannii simultaneously loaded into the chitosan microparticle for managing diabetic wound infection. The in vivo wound healing study demonstrated that the formulation produced maximum wound healing in 7 days. Additionally, it was found that the optimized formulation was nontoxic and safe for administration [7]. In another study, Ma et al. (2008) developed bacteriophage Felix O1 loaded chitosan alginate micro-formulations for oral delivery. The stability study suggested that free Bacteriophage was extremely sensitive to the acidic environment and ultimately died within 5 min at 3.7 pH. However, the Bacteriophage encapsulated within had good stability even at 2.4 pH.

Further, an in vitro release study demonstrated that the Bacteriophage was completely released in 6 h at pH 6.8 in simulated intestinal fluids. Hence, a developed formulation can deliver the encapsulated Bacteriophage to the gastrointestinal tract for therapeutic action [8]. Weppelmann et al. (2020) recently designed the chitosan microparticle for the vibriocidal activity toward MDR cholera infections. It was observed that chitosan microparticles of weight above 0.1 % efficiently killed the Vibrio cholera [9]. Moreover, Park et al. (2002) demonstrated that chitosan microparticles controlled the drug delivery of the loaded felodipine as the sodium TTP concentration increased, resulting in a slower release of the felodipine.

Additionally, as the molecular weight of the Chitosan decreased, a faster release rate was obtained [10]. A few years back, Sette-de-Souza et al. (2020) reported the antibacterial activity of Schinopsis brasiliensis extract loaded with chitosan microparticles for infectious disease. Microparticles were found to be safe based on the low hemolytic potential assay. Antimicrobial tests showed that microparticles could stop the growth of E. faecalis ATCC as well as the development of biofilms [11].

Comment 3: Methods: The whole genome of the phage should be sequenced.

Response: The author is thankful to the reviewer for the valuable suggestion that will enhance the overall quality of the research.

The Reviewer pointed out performing additional experiments for a whole genome of the phage. I agree these would be a good idea and further justify bacteriophage identification. The main reason I could not do additional experiments is that it would require additional funding that I don’t have and cannot acquire in a short time. Further, I would say that the current results of this manuscript are suitable for publication.

Comment 4: Methods: EOP should be performed.

Response: The author is thankful to the reviewer for the valuable suggestion.

All bacteria strains were provided by Prof. Gopal Nath, Department of Medical Microbiology, Institute of Medical Sciences (IMS), Banaras Hindu University, Varanasi (India). They will separately be going to publish the bacteria and bacteriophage interaction data, where the EOP study is also one of them. Therefore, we have not included the EOP study in this research paper. Further we mainly focused on bacteriophage delivery as personalized therapy. Nevertheless, we share the result data of EOP data to assure the reviewer.

We isolated the phage directly against the target bacteria (for personalized therapy). Therefore, our target strain is our propagation (host) strain (BHU/AB/39). However, calculated EOP by comparing it with the EOP of other patient-isolated strains are presented below.

Method: Efficiency of plating (EOP)

To determine the lytic productivity, bacteriophage samples were serially diluted. The successive dilutions ranging from 104–1010 PFU/mL were spot plated (20 μL) on the bacterial lawn of all Acinetobacter baumannii strains separately and incubated overnight at 37°C [6]. The titers were calculated, and EOP was determined as follows:

Phage productivity in terms of EOP values was classified as described by Mirzaei and Nilsson, 2015[7]:

High productivity          >0.5

Medium productivity     0.5–0.1

Low productivity        0.001–0.1

Inefficient productivity      <0.001

Result: Isolated Bacteriophage

S. No.

Bacteria

Source of isolation

EOP

Result

1.       

A. baumannii  (BHU/AB/41)

Human pus/swab

0.44±0.08

Medium productivity

2.       

A. baumannii  (BHU/AB/41)

Human sputum

0.74±0.31

High productivity  

3.       

A. baumannii  (BHU/AB/53)

Human urine

0.56±0.02

High productivity  

4.       

A. baumannii  (BHU/AB/76)

Human blood

0.71±0.04

High productivity  

5.       

A. baumannii  (BHU/AB/78)

Human pus/swab

0.03±0.00

Low productivity  

  1. Vashisth, M.; Jaglan, A.B.; Yashveer, S.; Sharma, P.; Bardajatya, P.; Virmani, N.; Bera, B.C.; Vaid, R.K.; Anand, T. Development and Evaluation of Bacteriophage Cocktail to Eradicate Biofilms Formed by an Extensively Drug-Resistant (XDR) Pseudomonas aeruginosa. Viruses 2023, 15, 427.
  2. Khan Mirzaei, M.; Nilsson, A.S. Isolation of phages for phage therapy: a comparison of spot tests and efficiency of plating analyses for determination of host range and efficacy. PloS one 2015, 10, e0118557, doi:10.1371/journal.pone.0118557.

Comment 5. Methods: Line 135: add India.

Response: As suggested by the reviewer, we have added India to the revised manuscript (Line 158 in the revised manuscript.

Comment 6. Methods: The author should briefly describe the DLAO method.

Response: The process of the Double layer agar overlay (DLAO) method, also called the double-layer agar (DLA) method, has already been explained in section 2.4.1, “Quantification” of the manuscript.

Comment 7: Results: Line 390: Phage should start with p.

Response: The rectification has been done as suggested in the revised manuscript.

Comment 8: Results: Fig 1(A) The enlarged figure is not corresponding with the plate. The author should check the ratio of the enlarged figure.

Response: The enlarged figure is the same as the plate. We have just changed the orientation of the enlarged image for better observation. In the revised manuscript, we have rectified the image. Below is the original image in an editable format, allowing the Reviewer to enlarge and rotate it for verification purposes.

Comment 9. Results: Fig 3(c) and Fig 4(b, d) The author should perform the statistical analysis.

Response: As suggested by the reviewer, we have added the statistical analysis for Fig 3(c), which is Fig 4 (c), in the revised manuscript.

Similarly, we also added the statistical analysis for Fig 4 (b, d) in the revised manuscript.

The revised manuscript has changed Fig 4 (b, d) to Fig 5 (b, d).

Figure 4. In vivo wound healing study on rat model where A) shows wound status on days 3, 7, 14, 21, and 28, (B) Histopathology examination of the skin of respected group, and (C) graphical data representation of % wound closure.

Figure 5. Ultrasound imaging of wound, where A) 3D ultrasound imaging of the Control group, 1% SSD (Marketed formulation) and treatment with BPABΦ-CHMPs-gel, B) Line graph indicating the change in wound 3D volume on the Y axis and time on X axis, C) Photoacoustic imaging of wound with D) graphical representation of changes in sO2 % of different groups on day 3, 14 and 21.

Comment 10. Results: The author should change the bar graph to the line graph to make it easier to understand. Statistical analysis should be performed.

Response: As suggested by the reviewer, we have changed the bar graph to the line graph and added the statistical analysis in the revised manuscript. Please see the Fig 3, line no 487.

Comment 11. Conclusion: The author performed the experiment against only A. baumannii, so the author should change the word from MDR bacteria to MDR A. baumannii.

Response: As suggested by the reviewer, we have changed MDR bacteria to MDR A. baumannii in the revised manuscript.

Comment 12. According to the stability results, the author should rewrite the stability of the entrapped phage (Lines 593-594).

Response: As per suggestion, we have rewritten the stability of entrapped phage. Please see the line no. 481-485.

References

  1. Ilomuanya, M.O.; Enwuru, N.V.; Adenokun, E.; Fatunmbi, A.; Adeluola, A.; Igwilo, C.I. Chitosan-Based Microparticle Encapsulated Acinetobacter baumannii Phage Cocktail in Hydrogel Matrix for the Management of Multidrug Resistant Chronic Wound Infection. Turkish journal of pharmaceutical sciences 2022, 19, 187-195, doi:10.4274/tjps.galenos.2021.72547.
  2. Ma, Y.; Pacan, J.C.; Wang, Q.; Xu, Y.; Huang, X.; Korenevsky, A.; Sabour, P.M. Microencapsulation of bacteriophage felix O1 into chitosan-alginate microspheres for oral delivery. Applied and environmental microbiology 2008, 74, 4799-4805, doi:10.1128/AEM.00246-08.
  3. Weppelmann, T.A.; Jeong, K.C.; Ali, A. Characterization of the Vibriocidal Activity of Chitosan Microparticles: A Potential Therapeutic Agent for Emerging Multidrug-Resistant Cholera Infections. ACS Applied Materials & Interfaces 2020, 12, 47278-47288, doi:10.1021/acsami.0c14313.
  4. Ko, J.A.; Park, H.J.; Hwang, S.J.; Park, J.B.; Lee, J.S. Preparation and characterization of chitosan microparticles intended for controlled drug delivery. Int J Pharm 2002, 249, 165-174, doi:10.1016/s0378-5173(02)00487-8.
  5. Sette-de-Souza, P.H.; de Santana, C.P.; Amaral-Machado, L.; Duarte, M.C.T.; de Medeiros, F.D.; Veras, G.; de Medeiros, A.C.D. Antimicrobial Activity of Schinopsis brasiliensis Engler Extract-Loaded Chitosan Microparticles in Oral Infectious Disease. AAPS PharmSciTech 2020, 21, 246, doi:10.1208/s12249-020-01786-x.

Reviewer 2 Report

11. Please identify the isolated A. baumannii and its antimicrobial resistance genes by molecular methods.

   2. Please ``an animal study`` added to title

3  3. Please specify the types and numbers of animals in the study exactly

4   4. Why has systemic antibiotic not been used in any of the animal groups to treat wound infection with A. baumannii?

5   5. In the method section, no clinical end point was explained, which ultimately made the results of the study ambiguous. Please explain about the clinical endpoints & responses and bring them in result section.

English writing needs to improve by native person.

Author Response

We sincerely thank the reviewer for his/her time and effort in providing valuable comments and suggestions. Corresponding changes have been made in red text in the revised version of the manuscript with track change mode, which is listed below point-to-point in bold text. The original comments are copied in italics text.

Comment 1: Identify the isolated A. baumannii and its antimicrobial resistance genes by molecular methods.

Authors response:  A. baumannii bacteria was provided by Prof. Gopal Nath, Department of Medical Microbiology, Institute of Medical Sciences (IMS), Banaras Hindu University, Varanasi (India). They will separately publish their molecular characterization and gene identification. Therefore, we cannot characterize bacteria further. However, we will update the information related to the bacteria once their data has been published.

Comment 2: Please ``an animal study`` added to the title.

Authors response:  Authors thanks to the reviewer for unique suggestion. But we are sorry for not considering the suggestions, as adding "an animal study" in the title will distort the aim of the study proposal. However, we have added "an animal study" as the subheading in the revised manuscript.

Comment 3: Please specify the types and numbers of animals in the study exactly.

Authors response:  A total of 15 (7 males and 8 females, randomly selected) Wistar rats (7-9 weeks, 200-250 grams) were used in the study.

Comment 4: Why have systemic antibiotics not been used in animal groups to treat wound infections with A. baumannii?

Authors response:  As we are more focused on the topical delivery system, we have compared our formulation with the marketed topical cream as the standard and have not considered any systemic antibiotics.

Comment 5: In the method section, no clinical endpoint was explained, which ultimately made the results of the study ambiguous. Please explain about the clinical endpoints & responses and bring them in result section.

Authors response:  The time required for complete wound re-epithelialization was considered the experiment's endpoint. The animals treated with formulation took 28 days for the eschar to come off without leaving any raw wounds. However, the USG imaging was unable to analyze the remained scars after 21 days, as they are very minute in size, and thus we performed the USG imaging after 21 days instead of 28 days. 

Comment 6: English writing needs to improve by a native person.

Authors response:  As per suggestion, we have improved the English writing in the revised manuscript.

Reviewer 3 Report

The manuscript presented by Dehari and colleagues focuses of the isolation, characterization, in vitro and in vivo evaluation of BP against MDR A. baumannii. The manuscript is well written and the scientific background as well as exprimental design had an overall merit. The materials and methods used are clearly stated and the results are supported by high quality images. Some amendments should be done before further proceeding. 

Author Response

We sincerely thank the Reviewer for the time, effort, valuable comments, and suggestions. Corresponding changes have been made in red text in the revised version of the manuscript with track change mode and are listed below point-to-point in bold text. The original comments are copied in italics text.

Comment 1: Line 1-2: change title as follows "A bacteriophage-based microgel for the treatment of multidrug-resistant Acinetobacter baumannii infection in burn wounds."

Authors response:  As suggested by the Reviewer, we have corrected the title "A bacteriophage-based microgel for the treatment of multidrug-resistant Acinetobacter baumannii infection in burn wounds" in the revised manuscript.

Comment 2: Abstract section, Lines 12-13: is one on the major pathogens isolated from burn wound infections.

Authors response: As per the suggestion we have corrected the sentence "is one on the major pathogens isolated from burn wound infections" in the revised manuscript.

Comment 3: Introduction section, Lines 31-33: change as follow "MDR A. baumannii bacterial infection is more common in patients with underlying diseases (e.g., ventilator-associated pneumonia, urinary tract infections, and bacteremia) or therapies that require invasive procedures."

Authors response: As per the suggestion we have corrected the line as "MDR A. baumannii bacterial infection is more common in patients with underlying diseases (e.g., ventilator-associated pneumonia, urinary tract infections, and bacteremia) or therapies that require invasive procedures" in the revised manuscript.

Comment 4: Introduction section, Lines 50-53: change as follow "It was discovered that the efficacy of bacteriophage treatment is mostly determined by bacteriophage concentration, implying that increasing MOI reduces bacterial population and hence enhances phage therapy."

Authors response: As per the suggestion we have corrected the line as "It was discovered that the efficacy of bacteriophage treatment is mostly determined by bacteriophage concentration, implying that increasing MOI reduces bacterial population and hence enhances phage therapy" in the revised manuscript (Line 48-51 in the revised manuscript).

Comment 5: Introduction section, Lines 69-86: this section needs revisions. Avoid the anticipation of particular materials or methods used (like SEPINEO) and summarize the aims of the study (and also the novelty of this study)

Authors response: The author is thankful for the suggestion, but the stated lines (lines 93-107: in the revised manuscript) describe a very important part of the research stating the novelty of the formulation which includes the usage of water-soluble grade chitosan which has been never used before, trehalose which improved the stability of the entrapped bacteriophage, and glycerol within the gel base enhancing the retention of bacteriophage within the wound site and increasing patient compliance. This novel combination of ingredients together makes the formulation stable and efficient for bacteriophage delivery.

Comment 6: Materials and Methods section, Isolation and characterization of bacteriophage subsection, Lines 147-151: have you followed a specific protocol for water filtration? (add a reference) Could you indicate the amount of filtered water?

Authors response: For isolation of bacteriophage and water filtration, the modified protocol of Singh A. et al 2022 was used, where 0.5 mL of filtrate was obtained from 2 mL of the collected water sample (Line 168-176 in the revised manuscript).

Comment 7: Materials and Methods section, Isolation and characterization of bacteriophage subsection, Line 158: which is the agar medium used? Have you performed triplicates?

Authors response: For isolation of bacteriophage, Muller Hinton agar + 1% Agar (Bacteriological grade) plate was used. The same plate was used as a base plate for double-layer agar overlay quantification. For upper-layer soft agar, Tryptic soya agar (0.6% agar) in distilled water was used.

Yes, all the quantification and characterization were performed in triplicate.

Comment 8: Materials and Methods section, burst size identification subsection, Lines 171-184: Have you performed triplicates?

Authors response: Yes, we have performed the studies in triplicate.

Comment 9: Materials and Methods section, thermal, pH, and UV stability study subsection, Lines 196-200: Have you performed triplicates?

Authors response: Yes, we have performed the study in triplicate.

Comment 10: Materials and methods section, antibiofilm assay subsection, Line 300: Why have you chosen this concentration?

Authors response: We have optimized the minimum biofilm eradication concentration (MBEC) of BPABΦ1-CHMPs by several trials, which was found to be greater than the MIC value that is 315.0±2.16 μg/mL. This was due to A. baumannii produced strong biofilms, and thus 2X concentration of MIC of BPABΦ1-CHMPs was selected.

Comment 11: Materials and methods section, in vivo study subsection, Line 321: could you add the animal's age and the number of animals recruited?

Authors response: We have provided a description of the total number of animals (15, 5 in each group) that are 7-9 weeks old in the revised manuscript (Lines 346-348 in the revised manuscript).

Comment 12: Materials and methods section, in vivo burn wound healing [..] subsection, Line 337-338: write the equation using the formatting style used above.

Authors response: We have formatted the formula in equation style in the revised manuscript as suggested (Line 364 in the revised manuscript)

Comment 13: Materials and methods section, Stability studies subsection: move this section prior to the in vivo subsection

Authors response: As per the suggestion, we have moved the stability section above in vivo study (animal study).

Comment 14: Discussion section, a general comment: you should include some results from other studies. In this form, your discussion is a repetition of concepts already stated in the results section. You need a comparison with existing literature.

Authors response: As per the suggestion, we have added the results from other studies and compared them with existing literature in the revised manuscript.

Round 2

Reviewer 1 Report

The manuscript has been revised according to the suggestions and comments.

Author Response

Comment: The manuscript has been revised according to the suggestions and comments.

Author's response: The authors are very thankful to the reviewer for finding the revised manuscript suitable for publication in the esteemed journal.

Reviewer 2 Report

1.       Please identify the isolated A. baumannii and its antimicrobial resistance genes by molecular methods.

2.       Please ``an animal study`` added to title

3.       Please specify the types and numbers of animals in the study

4.       Why has systemic antibiotic not been used in any of the animal groups to treat wound infection with A. baumannii?

5.       In the method section, no clinical end point was explained, which ultimately made the results of the study ambiguous. Please explain about the clinical endpoints & responses and bring them in result section.

the English writing needs to improve.

Author Response

Response to reviewer

It seems that the reviewer's comments for the 2nd revision are very much similar to that of the first revision.

We have already responded to point-to-point comments as provided in the first revision. 

Reviewer# 2 Report

  1. Identify the isolated A. baumannii and its antimicrobial resistance genes by molecular methods.

Authors response:  A. baumannii bacteria was provided by Prof. Gopal Nath, Department of Medical Microbiology, Institute of Medical Sciences (IMS), Banaras Hindu University, Varanasi (India). They will separately publish their molecular characterization and gene identification. Therefore, we cannot characterize bacteria further. However, we will update the bacteria information once their data has been published.

  1. Please ``an animal study`` added to title.

Authors response: As per the suggestion we have added the animal study in heading portion.

  1. Please specify the types and numbers of animals in the study exactly.

Authors response: A total of 15 (7 males and 8 females, randomly selected) Wistar rats (7-9 weeks, 200-250 grams) were used in the study.

  1. Why has systemic antibiotic not been used in any of the animal groups to treat wound infection with A. baumannii?

Authors response: We have prepared the topical formulation that’s why we used topical cream for standard.

  1. In the method section, no clinical end point was explained, which ultimately made the results of the study ambiguous. Please explain about the clinical endpoints & responses and bring them in result section

Authors response: The time required for complete re-epithelialization of wound was considered as endpoint of the experiment. In this experiment we found 28 days for the eschar to come off without leaving any raw wounds.in formulation group. But in USG was unable to 3D image of small remained scar, that’s why we have analysed it for 21 days.

  1. English writing needs to improve by native person.

Authors response: As per suggestion we have improved the english writing in whole manuscript.

Round 3

Reviewer 2 Report

Thanks to the authors team , unfortunately, most of the comments have not been considered and corrected.

the manuscript needs the minor English editing